# An Exploratory Quantitative Study of Factors Associated with Dissatisfaction with Japanese Healthcare among Highly Skilled Foreign Professionals Living in Japan

Tomoari Mori [1,2], Yoko Deasy [1], Katsumi Mori [3], Eisuke Nakazawa [3] and Akira Akabayashi [3,4,*]

1   Health Center, Okinawa Institute of Science and Technology Graduate University (OIST), Okinawa 904-0495, Japan
2   Department of General Medicine, University of Occupational & Environmental Health, Japan (UOEH), Fukuoka 807-8556, Japan
3   Department of Biomedical Ethics, Faculty of Medicine, University of Tokyo, Tokyo 113-0033, Japan
4   Division of Medical Ethics, School of Medicine, New York University, New York, NY 10016, USA
*   Correspondence: akirasan-tky@umin.ac.jp

**Abstract:** Background: To identify factors necessary for the proper inclusion of foreigners in Japanese healthcare, we conducted a survey to determine whether foreign residents, even those with high socioeconomic status, referred to as "Highly Skilled Foreign Professionals", experience difficulties when visiting medical institutions in Japan, using satisfaction level as an indicator. Method: A five-point Likert-scale, anonymous, online questionnaire was administered to faculty and doctoral students enrolled at the Okinawa Institute of Science and Technology Graduate University (OIST). The respondents' demographics and their opinions on what they found difficult or important during medical examinations, their impression of Japanese medical personnel, their requirements for language support and interpretation, and their opinions about local healthcare delivery systems were collected. The questionnaires were distributed and collected using Microsoft Forms. Results: Responses were obtained from 90 respondents (response rate: 20.7%). The percentage of respondents who were dissatisfied (bottom two of five Likert scales) with medical care in Japan was 23.4%. In univariate logistic regression analysis, 11 of 35 questions were significantly correlated with dissatisfaction with medical care ($p < 0.1$). Duration of stay, presence of family members living with the patient, satisfaction with life, and perceived usefulness of language support were negatively correlated with dissatisfaction with medical care, while communication problems with medical personnel, need for medical personnel to respect patients' culture, religion, and privacy, and difficulty in getting to medical facilities were positively correlated with dissatisfaction with medical care. No significant correlations were found with age, gender, or Japanese language level. Multiple logistic regression analysis showed that the presence of family members living with the subject (AOR = 0.092, $p = 0.010$), the desire for multilingual documentation (AOR = 0.177, $p = 0.046$), the physician's concern for the patient's culture and religion (AOR = 8.347, $p = 0.029$), and difficulty in communication with healthcare providers (AOR = 6.54, $p = 0.036$) were significantly correlated with overall dissatisfaction with medical care. Discussion: On average, the targeted Highly Skilled Foreign Professionals did not have strong levels of dissatisfaction with Japanese healthcare, but when they did have dissatisfaction, it was related to language barriers, lack of cultural and religious considerations, and difficulty in communication with healthcare providers. We believe that the results indicate the focus points of support necessary for the inclusion of foreigners. We also believe that the finding that family cohabitation is associated with satisfaction with medical care is a useful insight into effective reciprocal support on the part of patients.

**Keywords:** highly skilled foreign professionals; dissatisfaction with Japanese healthcare; language barriers; sociocultural aspects; Okinawa

## 1. Introduction

It is important to develop a system for accepting medical care for the rapidly increasing number of foreign visitors and residents in Japan in recent years. Based on such recognition at the level of government and medical associations, the Ministry of Health, Labor, and Welfare (MHLW) has developed a "Manual for Medical Institutions for the Acceptance of Foreign Patients" [1], and the Japan Medical Association has established a "Committee on Medical Countermeasures for Foreigners" [2], which also reports on COVID-19 infection-control measures. A few advanced medical institutions have obtained Japan Medical Services Accreditation for International Patients (JMIP) [3] and have established international departments specializing in treating foreigners. However, it is not easy for many busy Japanese medical institutions to provide adequate support for foreign patients. The MHLW has set a goal of establishing a system of medical institutions dedicated to foreigners in each secondary medical care area [4], but the actual status of the system varies greatly from municipality to municipality. In Japan's medical system, which is based on the premise of free access, it is expected that many foreigners will prefer to receive medical care at medical institutions that are easy to visit. While most travelers are in good health, and most of their medical visits are for sudden illness or injury, foreigners residing in Japan need to be provided with the same extensive and continuous medical care as Japanese people, not simply a stop-gap measure until they return to their home country.

Foreign residents are not uniform, and can vary widely in language, in culture and religion, as well as in their ideas about health literacy [5] and medical care. Regarding the experiences and difficulties of foreign residents in seeing a doctor, Teraoka et al. conducted a group interview of 22 foreigners and pointed out the lack of understanding of Japanese medical personnel regarding cultural and religious diversity, along with language barriers [6]. On the other hand, from the perspective of practical constraints, since it is difficult for Japanese medical institutions to respond to every individual cultural factor, to properly include non-Japanese individuals in medical care, it is vital to prevent mismatches between the needs of foreigners and the support system of Japanese medical institutions and to effectively allocate resources. To this end, it is important to gain knowledge about what the concerned individuals find difficult and what kind of support they need when they receive medical care.

To date, research on access to healthcare for foreigners has been mainly conducted in Europe and the United States and has focused on supporting immigrant communities of specific nationalities, mainly blue-collar workers and vulnerable members of society with low health literacy [7–14]. These studies have pointed out the difficulties in accessing medical care for non-English speakers and foreigners with low educational levels and socioeconomic status. It is not difficult to imagine that similarly socially vulnerable groups face difficulties in accessing medical care in Japan. However, there is a lack of information on whether white-collar foreigners, who are not socioeconomically vulnerable, face the same difficulties in accessing healthcare as immigrant foreigners in Japan, where a healthcare system for foreigners is being developed.

A typical example of such white-collar foreigners is Highly Skilled Foreign Professionals (HSFPs) [15], whom the Japanese government is trying to attract as a matter of policy. HSFPs possess a certain level of skills, are active internationally, and are generally expected to have stable socioeconomic conditions and a high level of health literacy. Although the Japanese insurance system is reported to be attractive to such foreigners [16], there are few studies that quantitatively investigate whether they experience any difficulties in receiving medical examinations. Therefore, we conducted an exploratory quantitative survey at the Okinawa Institute of Science and Technology Graduate University (OIST) [17], where a multinational community of HSFPs exists, to explore their evaluation of Japanese healthcare and its associated factors. Any difficulties they experience in seeing a doctor would indicate areas for improvement regarding medical care for foreigners in Japan. In this study, we used difficulty, importance, and satisfaction, as perceived by the subject, as indicators of access to healthcare. The survey items were based on themes extracted

from the results of a qualitative study using a thematic analysis of the same subject group's experience of receiving medical care in relation to difficulties in the experience of receiving medical care [18].

## 2. Participants and Methods

### 2.1. Survey

An online, anonymous, self-administered survey of OIST's resident foreign faculty and staff (435 as of April 2022) was conducted from June 2021 to April 2022. The inclusion criteria for the subjects were as follows: (1) holding an e-mail address issued by the university as a full-time employee or as a doctoral student; (2) understanding documents in English (all subjects understood English because it is the official language of the university); (3) foreign national, regardless of age, gender, or occupation, and (4) ability to understand and agree to the survey instructions. The survey request was sent via the campus intranet, and the target audience was asked to access the response page. The respondents were asked to read a description of the survey at the beginning, and if they understood the description and agreed to participate in the survey, they were considered to have given their consent to the survey by entering and submitting their responses.

### 2.2. Questionnaire

In designing the questions, we employed the themes identified in our earlier qualitative study [18] of the same foreign population at OIST. To ensure the objectivity of the items, we also checked citations [19–23] that investigated similar items as indicators of foreigners' cultural sensitivity and satisfaction with medical visits. This confirmed and reinforced the validity of the questionnaire items. The survey items included personal attributes such as age, gender, length of stay in Japan, Japanese language level, presence or absence of family members living with them, and region of origin, as well as questions on hospital visits and hospitalizations in Japan, acquaintances who could be asked to interpret when visiting a doctor, problems experienced when visiting a medical institution in Japan, what they thought was important about medical professionals, their impression of Japanese medical professionals, and their satisfaction with life and medical care in Japan. The respondents were also asked about language support at medical institutions, language support tools, the importance of interpreters, their family doctor, and their level of satisfaction with life and medical care in Japan. In total, 35 items for the questionnaires were developed, focusing on exploring the elements of difficulty, importance, and dissatisfaction related to medical visits. Five HSFPs were pre-tested, and we received feedback from them. Each question was answered either on a Yes/No scale or on a 5-point Likert scale, with the exception of the items related to personal attributes.

### 2.3. Medical Facility Environments near the OIST

The main island of Okinawa, the field of this study, is a remote island with a population of 1.29 million, and is neither urban nor completely rural. With its historical background of occupation by the U.S. military and the presence of U.S. military bases, it is also a relatively internationalized island with many foreign tourists. Below is a list of medical facilities by distance from OIST. Comparisons within Japan show that medical facilities are not as dense and convenient as in urban areas, but there are several medical facilities, ranging from clinics to general hospitals, with easy access by private car or public transportation. We believe that this survey allowed us to examine the general trends of medical facilities in the region, rather than the characteristics of a specific few.

500 m: Clinic (interpreter software available for foreigners to use)
3.5 km: Clinic (mainly for foreigners)
6.5 km: Clinics of Psychiatry, Internal Medicine, and Otolaryngology
18 km: General hospital (2.5 tier)
17 km: General hospital (tertiary)
25 km: General hospital (with International Medicine Department)

17.7 km: Radiotherapy and medical checkup clinic (outsourced medical checkups, foreign residents)

Roughly 20 other hospitals were also accessible to those in the southern part of the province, mainly because it is a small province.

### 2.4. On-Campus Medical Facilities

OIST Clinic: located on the university grounds.

Health Center: Open weekdays 9 am–5 pm (over-the-counter prescriptions can be dispensed). The clinic is open by appointment only for half a day in the morning or afternoon and first aid for sudden illnesses or emergencies is available in all departments on all days.

Health Center: Open weekdays 9 am–5 pm (over-the-counter prescriptions can be dispensed). The clinic is open by appointment only for half a day in the morning or afternoon, and first aid for sudden illnesses or emergencies is available in all departments on all days. Since health insurance is not available to cover treatment, employees pay the same amount of the cost (30%) as regular national insurance in Japan. The rest of the cost is supported by OIST.

### 2.5. Analysis

Responses to the question about satisfaction with medical care in Japan, using a 5-point scale ranging from "1: totally dissatisfied" to "5: very satisfied", were converted into binary values, with 1 and 2 as "dissatisfied" and 3, 4, and 5 as "satisfied". Logistic regression analysis was then conducted using this as the objective variable to examine the factors associated with feeling dissatisfied with Japanese medical care. Other items answered by the 5-item method were also converted to binary values and used after looking at the distribution of responses and ensuring that the number of people was not extremely skewed. Since there were many items to examine, univariate logistic regression analysis was first conducted using each item as an explanatory variable in order to narrow down the items that were expected to have some degree of association. Next, multiple logistic regression analysis was conducted using the items with $p < 0.1$ to obtain adjusted odds ratios (AOR) to examine the factors associated with dissatisfaction with medical care in Japan. Statistical analysis was performed using SAS version 9.4 (SAS Institute Inc., Cary, NC, USA). A value of $p < 0.05$ was considered statistically significant.

### 3. Results

The survey received responses from 90 participants, representing a response rate of 20.6% of the 435 enrolled as of April 2022.

Descriptive statistics on the demographics of the respondents are presented in Table 1. Respondents in their 30 s were the most common age group, followed by those in their 20 s and 40 s. Male and female respondents were almost equal in number, and the most common length of stay in Japan was 2 to 5 years. Almost the same number of respondents had family members living with them as those without, and more of them could not speak Japanese as well as those who could. The most frequent region of origin was Europe, followed by Asia and North America. The distribution of overall satisfaction with Japanese healthcare on a 5-point scale (1: very dissatisfied, 5: very satisfied) was 1 (5.6%), 2 (17.8%), 3 (32.2%), 4 (31.1%), and 5 (13.3%).

The results of the univariate logistic regression analysis on satisfaction with medical care in Japan are shown in Table 2. Crude odds ratio *p*-values lower than 0.1 were considered to be significant. Duration of stay in Japan, presence of family members living with the participant, and satisfaction with life in Japan were all correlated with satisfaction with Japanese medical care. Communication problems with medical professionals, and not knowing what to do in medical institutions, as experiences of visiting a medical institution, were also significantly correlated. Regarding the question about what patients want from Japanese healthcare professionals, respect for patients' culture and religion, and respect for

patients' privacy were significantly correlated. Regarding the question about impressions of Japanese health care providers, "the health care providers are friendly to my symptoms and treatment, and I get accurate answers to my questions" were significantly correlated. In the questions about useful language support, translation apps and bilingual documents were statistically significantly correlated. Age, gender, and Japanese language level were not significantly related to dissatisfaction with Japanese healthcare.

**Table 1.** Attributes of participants.

|  |  | Number of Persons (%) | |
|---|---|---|---|
| Age | 20–29 years old | 23 | (25.6%) |
|  | 30–39 years old | 42 | (46.7%) |
|  | 40–49 years old | 17 | (18.9%) |
|  | Over 50 years old | 8 | (8.9%) |
| Gender | Male | 42 | (46.7%) |
|  | Female | 43 | (47.8%) |
|  | Other/No answer | 5 | (5.6%) |
| Years in Japan | Less than 1 year | 9 | (10.0%) |
|  | 1–2 years | 17 | (18.9%) |
|  | 2–5 years | 35 | (38.9%) |
|  | 5–10 years | 15 | (16.7%) |
|  | 10 years or longer | 13 | (14.4%) |
|  | No response | 1 | (1.1%) |
| Japanese level | 1 (cannot speak) | 32 | (35.6%) |
|  | 2 | 31 | (34.4%) |
|  | 3 | 22 | (24.4%) |
|  | 4 | 3 | (3.3%) |
|  | 5 (can speak) | 2 | (2.2%) |
| Family living together | No | 46 | (51.1%) |
|  | Yes | 44 | (48.9%) |
| Birthplace | Asia | 21 | (23.3%) |
|  | Europe | 42 | (46.7%) |
|  | North America | 14 | (15.6%) |
|  | Latin America | 5 | (5.6%) |
|  | Other Regions | 8 | (8.9%) |
| Satisfaction with Japanese healthcare | 1 (very dissatisfied) | 5 | (5.6%) |
|  | 2 | 16 | (17.8%) |
|  | 3 | 29 | (32.2%) |
|  | 4 | 28 | (31.1%) |
|  | 5 (very satisfied) | 12 | (13.3%) |

**Table 2.** Factors associated with feeling dissatisfied with Japanese healthcare—univariate logistic regression analysis.

|  |  | Crude Odds Ratio (95% Confidence Interval) | | *p*-Value |
|---|---|---|---|---|
| Age | Over 40 (vs. Under 40) | 0.604 | (0.208, 1.752) | 0.353 |
| Gender | Female (vs. Male) | 1.645 | (0.594, 4.555) | 0.338 |
|  | Other (vs. Male) | 1.062 | (0.104, 10.840) | 0.959 |
| Length of stay | More than 5 years (vs. Less than 5 years) | 0.170 | (0.037, 0.791) | 0.024 |
| Japanese (language) | Can speak (vs. Cannot) | 0.668 | (0.217, 2.056) | 0.481 |
| Family living together | Yes (vs. No) | 0.240 | (0.079, 0.730) | 0.012 |
| Satisfaction with life in Japan | Unsatisfactory (vs. Satisfactory) | 11.333 | (1.112, 115.559) | 0.040 |

**Table 2.** *Cont.*

| | | Crude Odds Ratio (95% Confidence Interval) | | *p*-Value |
|---|---|---|---|---|
| Regular hospital visits | Yes (vs. No) | 0.938 | (0.333, 2.636) | 0.903 |
| Hospitalization in Japan | Yes (vs. No) | 0.414 | (0.125, 1.368) | 0.148 |
| An acquaintance who can interpret for you when you visit a medical facility | Yes or Probably (vs. No) | 1.114 | (0.419, 2.966) | 0.828 |
| Communication problems with healthcare professionals | Often (vs. Never) | 5.525 | (1.683, 18.137) | 0.005 |
| Not sure which medical facility to visit | Often (vs. Never) | 0.772 | (0.288, 2.067) | 0.607 |
| When you visit a medical facility, you do not know what to do | Often (vs. Never) | 3.050 | (0.919, 10.124) | 0.069 |
| The physician must speak English | Very important (vs. Not very important) | 1.611 | (0.603, 4.300) | 0.341 |
| Medical technology in the health professions | Very important (vs. Not very important) | 1.129 | (0.362, 3.528) | 0.834 |
| Adequate explanation by the healthcare professional to the patient | Very important (vs. Not very important) | 2.205 | (0.456, 10.675) | 0.326 |
| Healthcare professionals' consideration of patients' culture and religion | Very important (vs. Not very important) | 3.011 | (1.083, 8.374) | 0.035 |
| Healthcare professionals' concern for patient privacy | Very important (vs. Not very important) | 2.574 | (0.894, 7.412) | 0.080 |
| Japanese medical professionals are friendly and easy to talk to | Agree (vs. Disagree) | 1.989 | (0.599, 6.612) | 0.262 |
| Discriminated against by Japanese medical professionals because of being a foreigner | Agree (vs. Disagree) | 1.571 | (0.588, 4.200) | 0.368 |
| Japanese medical professionals were accommodating to my symptoms and treatment | Agree (vs. Disagree) | 0.328 | (0.119, 0.902) | 0.031 |
| The Japanese medical professionals answered my questions exactly as I asked them | Agree (vs. Disagree) | 0.257 | (0.089, 0.744) | 0.012 |
| Foreign language support should be provided in medical facilities | Agree (vs. Disagree) | 1.631 | (0.586, 4.541) | 0.349 |
| Languages other than English should be supported | Yes (vs. No) | 1.000 | (0.355, 2.818) | 1.000 |
| Professional translation | Useful (vs. Unhelpful) | 0.957 | (0.355, 2.579) | 0.931 |
| Translation apps | Useful (vs. Unhelpful) | 0.316 | (0.109, 0.912) | 0.033 |
| Hospital medical interpreters | Useful (vs. Unhelpful) | 0.479 | (0.173, 1.324) | 0.156 |
| Bilingual display of reception form and medical questionnaire | Useful (vs. Unhelpful) | 0.262 | (0.081, 0.844) | 0.025 |
| Accuracy of the translated content (when hiring an interpreter) | Important (vs. Not important) | 0.694 | (0.230, 2.099) | 0.518 |
| Help with reception and payment (when hiring an interpreter) | Important (vs. Not important) | 0.770 | (0.288, 2.058) | 0.602 |
| Cost (when hiring an interpreter) | Important (vs. Not important) | 0.934 | (0.342, 2.555) | 0.895 |
| I can ask for it ASAP (when hiring an interpreter) | Important (vs. Not important) | 0.969 | (0.328, 2.868) | 0.955 |
| Information on choosing a medical institution | Sufficient (vs. Insufficient) | 0.805 | (0.301, 2.153) | 0.666 |
| Medical coordinator | Necessary (vs. Not necessary) | 0.742 | (0.279, 1.974) | 0.550 |
| The community needs a primary-care physician | Agree (vs. Disagree) | 1.286 | (0.460, 3.594) | 0.632 |
| Family doctor in Japan | Yes (vs. No) | 0.445 | (0.092, 2.157) | 0.315 |

Table 3 shows the results of the multiple logistic regression analysis. The analysis used feeling dissatisfied with medical care in Japan as the objective variable and, in addition to items for which the *p*-value of the crude odds ratio were less than 0.1 by univariate logistic regression analysis, age and sex were used as explanatory variables as adjustment factors. According to the results of the univariate logistic regression analysis, the *p*-value of the crude odds ratio for satisfaction with life in Japan was also less than 0.1, and was

an item that needed to be added as an explanatory variable. However, adding it to the explanatory variables in the multiple logistic regression analysis resulted in extremely large standard errors in the estimates and may have made the solution unstable. For this reason, satisfaction with life in Japan was excluded from the explanatory variables.

**Table 3.** Factors associated with feeling dissatisfied with Japanese healthcare—multiple logistic regression analysis.

| | | Adjusted Odds Ratio (95% Confidence Interval) | | *p*-Value |
|---|---|---|---|---|
| Length of stay | More than 5 years (vs. Less than 5 years) | 0.462 | (0.066, 3.243) | 0.438 |
| Family living together | Yes (vs. No) | 0.092 | (0.015, 0.560) | 0.010 |
| Communication problems with healthcare professionals | Often (vs. Never) | 6.540 | (1.127, 37.950) | 0.036 |
| When you visit a medical facility, you do not know what to do | Often (vs. Never) | 0.866 | (0.107, 6.978) | 0.892 |
| Healthcare professionals' consideration of patients' culture and religion | Very important (vs. Not very important) | 8.347 | (1.242, 56.101) | 0.029 |
| Healthcare professionals' concern for patient privacy | Very important (vs. Not very important) | 0.854 | (0.146, 5.006) | 0.861 |
| Japanese medical professionals were accommodating to my symptoms and treatment | Agree (vs. Disagree) | 0.715 | (0.129, 3.959) | 0.701 |
| The Japanese medical professionals answered my questions exactly as I asked them | Agree (vs. Disagree) | 0.232 | (0.039, 1.391) | 0.110 |
| Translation apps | Useful (vs. Unhelpful) | 0.617 | (0.140, 2.708) | 0.522 |
| Bilingual display of reception form and medical questionnaire | Useful (vs. Unhelpful) | 0.177 | (0.032, 0.966) | 0.046 |

Age and gender as adjustment factors. Hosmer–Lemeshow goodness-of-fit test: *p* = 0.571.

The presence of family members living together was statistically significant (AOR = 0.092, *p* = 0.010). Those with family members living with them were less dissatisfied with medical care in Japan. The next largest influence was health professionals' consideration of patients' culture and religion (AOR = 8.347, *p* = 0.029). Those who thought it was very important for health professionals to pay attention to the culture and religion of their patients were more dissatisfied with Japanese healthcare. The results also showed that the respondents were dissatisfied with Japanese medical care if they had many communication problems with medical professionals, and were not dissatisfied with Japanese medical care if they thought it would be helpful if the reception form and medical questionnaire were written in two languages.

## 4. Discussion

This study examined issues related to the access to and inclusion of healthcare in the multinational community of HSFP, which is considered to have high socioeconomic status and health literacy, unlike immigrants of certain nationalities who have traditionally been perceived as socially vulnerable, using a subjective measure of the satisfaction of the parties involved. The results showed that, on average, the targeted HSFPs did not have major dissatisfaction with medical care in Japan, but when patients were dissatisfied with medical care, it was correlated with certain aspects such as difficulty in communicating with medical personnel, the attitude of medical personnel toward patients and patient care, and preference for language support. No apparent relationships were found with subjects' age, gender, or Japanese language level.

After adjusting for covariates using multiple logistic regression analysis, four items were obtained as significant factors. Of these, the result "dissatisfaction with Japanese medical care when there are many communication problems with medical personnel" indicated that the subjects associated communication difficulties with low satisfaction with medical care in Japan, although the causal relationship was unclear. The result "subjects who feel more favorably about bi-lingual documents are more satisfied with Japanese medical care" was quite difficult to interpret, but it may be that the participants focused on the language issue because they did not perceive any other major problems. The result "the more important the consideration of the patient's culture and religion by the healthcare provider, the more dissatisfied the patient is with the medical care" can be interpreted in two main ways. The first is that individual differences in the ability of Japanese healthcare professionals to deal with cultural and religious diversity may be the cause of differences in patient satisfaction. The second possibility is that the importance of consideration for cultural and social norms on the part of patients differs depending on the individual and their cultural background, and this may cause differences in satisfaction levels. Which of the above is related to the difference in satisfaction cannot be identified from this survey and will need to be clarified in future surveys. Even with a survey method such as this, if the sample size is large enough and stratified analysis by nationality and other basic attributes of the subjects is possible, some inferences may be drawn.

Although many studies of cross-cultural communication in healthcare, mainly in Europe and the United States, have discussed the importance of cross-cultural sensitivity, there is still no clear evidence on results such as patient outcomes [22,24–27]. However, in Japan, studies of nurses have reported that nurses avoid engaging with foreign patients by not listening sufficiently to their complaints, walking away from them, and providing only minimal care [28–31]. In both of the aforementioned situations, it is important for Japanese healthcare professionals to aim for more appropriate cross-cultural understanding as a necessary part of Japanese healthcare in order to properly accommodate foreigners from diverse backgrounds.

Regarding the result "medical satisfaction is higher when patients have family members living with them" as a factor on the patient side, it is speculated that, in addition to direct family support during medical visits, other confounding factors, such as satisfaction with life, may be involved. However, it is not possible to identify further causes or infer causal relationships from this study. Prior research has indicated that marital status affects the degree of independence in receiving medical care [13] and, more generally, the availability of family support acts as a buffer [32–34]. If further research is conducted to clarify the role of the presence of family members living with the patient and how it relates to satisfaction with medical care, it may be possible to consider the inclusion of foreigners in medical care from the perspective of social support, other than support provided by medical care at the time of medical visits.

Among the items for which the odds ratios were large but did not reach statistical significance, such as the consideration of patient privacy by healthcare providers, the reason may be that there were large differences in opinion among the participants, which is why the confidence intervals were large and not significant. For such items, if a sufficient number of subjects could be secured, and if the subjects could be properly stratified, it might be possible to identify some groups with strong opinions and wishes. This will be a consideration for future large-scale surveys. However, we must also recognize that it is dangerous to oversimplify the diversity of foreigners and to try to understand them only in terms of their nationality and the other basic attributes of the group to which they belong. In a qualitative survey of the same target population that we conducted prior to this study, HSFPs expressed negative opinions about stereotypical cultural biases [18]. To be inclusive of foreigners from diverse backgrounds, it is important to take a case-by-case approach with patient-centered care in mind, considering that individual opinions are diverse [35].

If we are to consider the inclusion of foreign patients in healthcare from a larger perspective, we need to be aware of the limitations of considering patient inclusion only in

terms of expressed dissatisfaction. Even in difficult situations, there may be occasions on which patients are unaware of their difficulties due, for example, to adaptation preferences, or on which they are unable to actively express their difficulties as complaints. This study could not grasp the subjects' difficulties in such situations. More multifaceted research is needed for Japanese healthcare workers to better understand the needs of diverse foreign populations in the future. From the viewpoint of providing appropriate medical care for foreigners, the study indicates that, at the very least, medical personnel who have frequent contact with foreigners need to be aware of what is troubling their patients and be flexible in dealing with a variety of patients. However, there is a limit to what busy frontline medical personnel can achieve solely through their own efforts. As the next step after the development of manuals, we should recognize the importance of organizational support for healthcare providers [36]. To improve the current cross-cultural unfamiliarity in Japan, it is desirable to develop educational programs for healthcare professionals that allow them to respond on a case-by-case basis, if possible [37].

*Limitations*

The ultimate goal of this study was to contribute to the inclusion of foreigners in healthcare. However, the actual study was a small-scale exploratory study conducted at a single university in Okinawa, where English is the official language, with a specific group of foreigners selected as the subjects. Therefore, the generalizability of the results is severely limited. The response rate was 20.7%, and the number of respondents was very small (90). In addition to the limitations of generalization, self-selection bias, in which only subjects with some feelings toward medical care responded, was expected, but the extent of this bias is difficult to assess. It was also not possible to conduct a more detailed analysis by stratifying the subjects by a particular attribute or degree of dissatisfaction. The indicators used to evaluate access to medical care were perceived difficulty, importance, and satisfaction. Therefore, we were not able to evaluate difficulties of which the participants were not aware or that they did not express. To overcome these limitations, a larger-scale study using a variety of foreigners in other regions as subjects would be desirable.

**5. Conclusions**

The subjects of this study, Highly Skilled Foreign Professionals, were generally satisfied with medical care in Japan. However, when they were dissatisfied with medical care, this was associated with dissatisfaction with communication with medical personnel and a desire for a consideration of their culture and religion by healthcare providers. In addition, the strength of the desire for multilingual medical interview forms and other documents and the presence of family members living with the patient were associated with high satisfaction with medical care. Further understanding of the reasons for these results may provide clues as to the appropriate inclusion of target foreigners; however, considering the diversity of values and opinions among the subjects, a stereotypical approach that groups all foreigners together may not be desirable. To better include a diverse range of foreign residents, it is necessary to conduct surveys on a large scale and in a diverse field that includes socially vulnerable foreign residents, and to understand the difficulties of which they may not be aware or that they may not be able to express as complaints. We also need to deliver these findings to medical professionals in the field and facilitate the development of a system to support them.

**Author Contributions:** Conceptualization, T.M. and Y.D.; methodology, T.M., Y.D. and K.M.; validation, E.N.; formal analysis, K.M.; writing—original draft preparation, T.M.; writing—review and editing, E.N. and A.A.; project administration, A.A. All authors have read and agreed to the published version of the manuscript.

**Funding:** This study was supported by a Grant-in-Aid for Scientific Research (grant no. JP20K18947) from the Japan Society for the Promotion of Science.

**Institutional Review Board Statement:** This study was conducted after ethical review and consent (review number: HSR-2020-017) by the Okinawa Institute of Science and Technology Graduate University Human Subjects Research Review Committee.

**Informed Consent Statement:** Not applicable.

**Data Availability Statement:** Not applicable.

**Conflicts of Interest:** The authors declare no conflict of interest.

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
