# Peer review of "An Exploratory Quantitative Study of Factors Associated with Dissatisfaction with Japanese Healthcare among Highly Skilled Foreign Professionals Living in Japan"

_2673-8430, doi:10.3390/biomed2040034_

Round 1

Reviewer 1 Report

On a general basis I’ve found the aim of the study interesting. Keep the health care system more inclusive and equal for every citizen local or foreign is very important, like understanding the main challenges and difficulties in attaining this objective. From a methodological point of view the information regarding the location of medical facilities environments should have to be provided with some more information and explanation regarding why you report it and how it could be helpful in discussing the results. Furthermore, you should have to provide more information about the study design and about the sampling process. I’ve found a lack of bibliography specifically when you write about the determination of the items that you’ve used in the study. You should have to cite the studies from which your items have been determined. As the nature of the study is cross-sectional, you cannot do inferences regarding a causality relationship (as you suggest in the conclusions) between communication problems and dissatisfaction. The article's topic is particularly interesting and could represent a little aid for future studies due to the final sample size. However, I would suggest meticulously reviewing the English to make the work more understandable. I would also suggest avoiding any causal inference due to the study's nature and the data's quality. Additionally, the authors could consider other types of a statistical analysis based on the sample size. 

I have difficulty defining the relationship between the analyses described in the paper and the final conclusions, which I would suggest should be revised and modified. In conclusion, I think that, after the proposed revisions, the paper can be published.

In what follow, I put some considerations:

Abstract 

19: Please, add information about the 90 faculty;

35-36: Please, pay attention to this sentence; I think there is a typo “Second, those who think it is very important 34 for medical professionals to pay attention to the culture and religion of their patients are more dissatisfied with 35 Japanese medical care than those who think it is very important.”

Introduction 

The introduction is clear and concise; however, I suggest that the author have to better focalize and describe the aim of their study in the final part of it. 

Participants and Methods 

105-106: Authors specified that “These items are determined from search of previous studies, discussion among research group teams.” Based on this, I ask the authors whether they report the same items or whether they were inspired by the “previous studies” in order to formulate them. In the first case, I suggest the authors to include references. 

112: Please, expand and describe better the section 2.3. More in detail, please better specify why the clinics were listed in order of distance. 

133: With respect to statistical analysis, both the univariate and multiple logistic regressions present too wide CI (e.g., “Satisfaction with life in Japan”: 11.333 (1.112;115.559)); this might depend on having conducted the logistic regression analysis on a small sample. Please, consider to do a correlation analysis considering the independent variables as continuous variables. 

134: Please, better describe the Likert-scale used inserting the label for each value. 

137-139: The sentence is quite understandable. Please, better describe “The other five responses […]”.

158, 169 and 195: The tables’ structure is not clear; specifically, regarding independent variables, it is not clear which is the reference category. I suggest to restructured it. For example, “Male vs Female” instead of “Male/Female”. 

169: Please specify why you compared “Other” category only with “Male” category in the univariate analysis. 

Discussion and conclusion 

I believe that the discussion is not very informative; moreover, the novel findings are established on the basis of statistical analysis that should be revised. Furthermore, considering the study’s design, it is not possible talking about cause-effect relationships, but associations. 

270-271: Pay attention, there is a typo. 

Author Response

Reply to Rev 1 Comments and Suggestions for Authors

On a general basis I’ve found the aim of the study interesting. Keep the health care system more inclusive and equal for every citizen local or foreign is very important, like understanding the main challenges and difficulties in attaining this objective. From a methodological point of view the information regarding the location of medical facilities environments should have to be provided with some more information and explanation regarding why you report it and how it could be helpful in discussing the results. Furthermore, you should have to provide more information about the study design and about the sampling process. I’ve found a lack of bibliography specifically when you write about the determination of the items that you’ve used in the study. You should have to cite the studies from which your items have been determined. As the nature of the study is cross-sectional, you cannot do inferences regarding a causality relationship (as you suggest in the conclusions) between communication problems and dissatisfaction. The article's topic is particularly interesting and could represent a little aid for future studies due to the final sample size. However, I would suggest meticulously reviewing the English to make the work more understandable. I would also suggest avoiding any causal inference due to the study's nature and the data's quality. Additionally, the authors could consider other types of a statistical analysis based on the sample size. 

I have difficulty defining the relationship between the analyses described in the paper and the final conclusions, which I would suggest should be revised and modified. In conclusion, I think that, after the proposed revisions, the paper can be published.

In what follow, I put some considerations:

Abstract 

19: Please, add information about the 90 faculty;

Thank you very much for pointing this out. The description was misleading as if there were other staff members apart from the 90 faculty members. The correct number is 90 respondents out of the total number of foreign faculty members and doctoral students. We have completely revised the description in the abstract, along with our response to the second reviewer's comments. For the additional information about faculty members, only the basic information presented in Table 1 was collected to ensure the anonymity of respondents from the perspective of privacy protection. While it is possible to provide basic attribute information for the entire faculty and staff of the university, it is unfortunately difficult to provide any more information on participants in this study.

35-36: Please, pay attention to this sentence; I think there is a typo “Second, those who think it is very important 34 for medical professionals to pay attention to the culture and religion of their patients are more dissatisfied with 35 Japanese medical care than those who think it is very important.”

We apologize for the basic mistake. We have corrected it. We have also carefully reviewed the entire manuscript and corrected other errors as well.

Introduction 

The introduction is clear and concise; however, I suggest that the author have to better focalize and describe the aim of their study in the final part of it. 

Thank you very much for your valuable feedback. As you pointed out, it was not clearly explained why we targeted "Highly Skilled Foreign Professionals". So, we have made the following additions and corrections to the last part of the background.

“To date, research on access to healthcare for foreigners has been mainly conducted in Europe and the United States and has focused on supporting immigrant communities of specific nationalities, mainly blue-collar workers and vulnerable members of society with low health literacy. These studies have pointed out the difficulties in accessing medical care for non-English speakers and foreigners with low educational levels and socioeconomic status. It is not difficult to imagine that similarly socially vulnerable groups face difficulties in accessing medical care in Japan. However, there is a lack of information on whether white-collar foreigners who are not socioeconomically vulnerable face the same difficulties in accessing healthcare as immigrant foreigners in Japan, where a healthcare system for foreigners is being developed.

A typical example of such white-collar foreigners is Highly Skilled Foreign Professionals (HSFPs), whom the Japanese government is trying to attract as a matter of policy. HSFPs possess a certain level of skills, are active internationally, and are generally expected to have stable socioeconomic conditions and a high level of health literacy. Although the Japanese insurance system is reported to be attractive to such foreigners, there are few studies that quantitatively investigate whether they experience any difficulties in receiving medical examinations. Therefore, we conducted an exploratory quantitative survey at the Okinawa Institute of Science and Technology Graduate University (OIST), where a multinational community of HSFPs exists, to explore their evaluation of Japanese healthcare and its associated factors. Any difficulties they experience in seeing a doctor would indicate areas for improvement regarding medical care for foreigners in Japan. In this study, we used difficulty, importance, and satisfaction, as perceived by the subject, as indicators of access to healthcare. The survey items were based on themes extracted from the results of a qualitative study using a thematic analysis of the same subject group’s experience of receiving medical care in relation to difficulties in the experience of receiving medical care.“

Participants and Methods 

105-106: Authors specified that “These items are determined from search of previous studies, discussion among research group teams.” Based on this, I ask the authors whether they report the same items or whether they were inspired by the “previous studies” in order to formulate them. In the first case, I suggest the authors to include references.

Although we could not describe it within the manuscript because it was not yet published at the time of this paper’s submission, the rationale for the choice of the items is precisely based primarily on the results obtained in our earlier qualitative study. We also checked prior studies for similar questions, although they did not use the same items, to ensure the validity of our questionnaire. This has been added to the text. We have presented the references as references cited and also modified the description as follows. 

“In designing the questions, we employed the themes identified in our earlier qualitative study of the same foreign population at OIST. To ensure the objectivity of the items, we also checked the citations that investigated similar items as indicators of foreigners’ cultural sensitivity and satisfaction with medical visits. This confirmed and reinforced the validity of the questionnaire items.”

112: Please, expand and describe better the section 2.3. More in detail, please better specify why the clinics were listed in order of distance. 

Thank you very much for your important feedback. We have presented the distance information from OIST to each medical facility to indicate that medical facilities in the study area are not very sparse, that subjects have a good choice of medical facilities, and that this study is not a survey of the characteristics of a specific few medical facilities. The following text was added to supplement the description.

“The main island of Okinawa, the field of this study, is a remote island with a population of 1.29 million and is neither urban nor completely rural. With its historical background of occupation by the U.S. military and the presence of U.S. military bases, it is also a relatively internationalized island with many foreign tourists. Below is a list of medical facilities by distance from OIST. Comparisons within Japan show that medical facilities are not as dense and convenient as in urban areas, but there are several medical facilities, ranging from clinics to general hospitals, with easy access by private car or public transportation. We believe that this survey allowed us to examine the general trends of medical facilities in the region, rather than the characteristics of a specific few.”

133: With respect to statistical analysis, both the univariate and multiple logistic regressions present too wide CI (e.g., “Satisfaction with life in Japan”: 11.333 (1.112;115.559)); this might depend on having conducted the logistic regression analysis on a small sample. Please, consider to do a correlation analysis considering the independent variables as continuous variables. 

Thank you for pointing that out. It is true that the confidence intervals for some of the odds ratios do appear to be large. However, in logistic regression analysis, it is not uncommon for the upper end of the confidence interval to be large because the model parameters are exponentially transformed to obtain the odds ratio estimates and confidence intervals. This tendency is especially noticeable when the odds ratio estimates are large. In fact, the standard errors are not extremely large for the parameters of the model. For example, in the case of "Satisfaction with life in Japan," the standard error is 1.1847 for an estimated parameter value of 2.4277. In univariate analysis, n=90 is not small for 1 degree of freedom in the model, and we believe that a stable solution is obtained. In the multivariate analysis, n=88 for 13 degrees of freedom of the model, which may be influenced by the small number of n. However, as noted in the Results section, items with extremely large standard errors were excluded from the explanatory variables in the process of analysis. Therefore, we believe that the resulting multivariate analysis results are also stable.

134: Please, better describe the Likert-scale used inserting the label for each value. 

The questionnaire showed numbers 1-5 as options, and only labeled 1 as "totally unsatisfactory" and 5 as "very satisfied," while 2 through 4 were not specifically labeled. This resulted in the wording as shown in the text.

137-139: The sentence is quite understandable. Please, better describe “The other five responses […]”.

Thank you for pointing this out. Since the wording was difficult to understand, we have revised the text as follows.

The first sentence:

“Regarding the level of satisfaction with medical care in Japan, the five items from "1. totally dissatisfied" to "5. very satisfied" were converted into binary values, with 1 and 2 as "dissatisfied" and 3, 4, and 5 as "satisfied.”

The third sentence:

“The other items answered by the 5-point scale were also converted to binary values after looking at the distribution of responses and making sure the number of people was not extremely unbalanced.”

158, 169 and 195: The tables’ structure is not clear; specifically, regarding independent variables, it is not clear which is the reference category. I suggest to restructured it. For example, “Male vs Female” instead of “Male/Female”. 

Thank you for your instruction. The table has been modified as suggested to clarify the reference categories.

169: Please specify why you compared “Other” category only with “Male” category in the univariate analysis. 

In general, when using qualitative explanatory variables in logistic regression analysis, one category is used as the reference category and the odds ratio of the other category to the reference category is shown. This is a natural presentation of the results from the parametrization of qualitative variables in regression models. For this reason, in Table 2, Male is used as the reference category and the odds ratios of Female and Other to Male are displayed.

Discussion and conclusion 

I believe that the discussion is not very informative; moreover, the novel findings are established on the basis of statistical analysis that should be revised. Furthermore, considering the study’s design, it is not possible talking about cause-effect relationships, but associations. 

We really appreciate your detailed comments. We have made significant revisions in response to the comments of the first and second reviewers. Please take a moment to review the revised version.

270-271: Pay attention, there is a typo. 

Thank you very much. We have made the correction.

Once again thank you for reading our manuscript in detail.

Reviewer 2 Report

The article presents an interesting topic. It is a very interesting article, it is coherent in its logic, structure, and way of organizing ideas. The main recommendations are set out below:

“the majority of foreign residents are foreign residents” clarify idea

In the abstract the objective of the article should be evidenced more clearly, specify to a greater degree

The components that make up the abstract must be clearly identified, for example, the methodological basis is not stated precisely, as well as the results; Likewise, some conclusive idea should be pointed out. (I suggest that it be organized in a structured way, clearly show introductory idea, objective, methodology, results, discussion, conclusions and complement the article's contributions to the advancement of knowledge of the object)

“To this end, educational methods need to be developed.” this idea appears isolated, show a greater relationship with the content of the abstract.

In the introduction, the objectives of the article, research questions should be evidenced. It is important to infer how this research contributes to the advancement of science, its contributions and social impact, identifying direct and indirect beneficiaries, as well as the relevance of this research within the framework of emerging paradigms that explain the educational dynamic.

The theoretical and empirical gaps of the situation under study should be evidenced to a greater degree, as well as the descriptors associated with the main variables studied; the variables and their dimensions should be described with greater emphasis. It is necessary to strengthen the citations of recent scientific literature that allow contrasting the descriptors of reality and show the state of knowledge related to the situation under study.

The situation under study, as well as the associated descriptors, the importance and relevance of the topic, the sense of contextualization in the region studied must be strengthened in their description; Likewise, the citation system of recent scientific literature related to the subject should be strengthened.

It is important to propose a deductive route; it is necessary that the implications of the problem situation be described on a macro, meso and micro plane; situation that must be described in greater depth because it must be better argued; the problem must start from a more general scope, before falling into the variables of the investigation; that is, they should begin by presenting descriptors of the macro; then locate yourself in the meso and micro plane,

The scope of the applied survey should be evidenced to a greater degree as a basis for the generalization of the results.

I suggest reviewing the logic and systematization of the methodological design, for this I recommend considering the following elements:

It is necessary to identify and justify the typology of the article.  Is important  specify the predominant research approach, as well as the type of design. It would have to be determined if it applies: the inclusion and exclusion criteria in the sample must be specified.  Likewise, the information gathering techniques and instruments must be specified; instruments for each of the identified samples; criteria of validity and reliability of the instruments. Likewise, it is necessary to specify the information processing techniques to be obtained. The procedural systematization must be reviewed organize the research stages, in correspondence with the objectives of the article, type of research according to knowledge to be produced and expected products. Provide greater evidence of the operationalization of its variables, main dimensions and indicators that allow measuring the behavior of its fundamental variables.

The profile of the subjects that make up each group, their characteristics, is not clear; should systematize in a more organized way.

Systematize the methodology in a more organized, logical and coherent way

Review the way to organize the results; there must be a better correspondence with the procedural systematization that is necessary to declare in the methodology, that is, how each component of the design leads to the different sections that are presented in the results. Review the correspondence between the methodological systematization and the systematization of the results, attend to variables, dimensions and indicators and their operationalization.

There is no analysis of the results, it is necessary to show: Organize the results in such a way that they are evident in relation to the objectives; there is no evidence of contrast between the objectives - supporting theory - meaning of the data itself - argument of the researchers.

Where is the argumentation or counter-argumentation with the supporting scientific literature?

It is necessary to strengthen the citations of recent scientific literature, citations of contrasting literature must be updated.

the conclusions show correspondence with the identified objectives; demonstrate possibilities of generalization of the research to contexts with similar characteristics. Conclusions must transcend results. Conclusions should be more precise

I suggest organizing the conclusions in a more precise way, as conclusive ideas in correspondence with the objectives, variables and dimensions of the study, the results should be transcended.

Author Response

The article presents an interesting topic. It is a very interesting article, it is coherent in its logic, structure, and way of organizing ideas. The main recommendations are set out below:

“the majority of foreign residents are foreign residents” clarify idea

Thank you very much for pointing this out. It was a simple writing error. We have also received a separate request to review the entire abstract, so we have revised the description in its entirety.

In the abstract the objective of the article should be evidenced more clearly, specify to a greater degree

We understand your point of view. Based on the advice given by you two reviewers, we have revised the abstract in its entirety. We would appreciate your review.

The components that make up the abstract must be clearly identified, for example, the methodological basis is not stated precisely, as well as the results; Likewise, some conclusive idea should be pointed out. (I suggest that it be organized in a structured way, clearly show introductory idea, objective, methodology, results, discussion, conclusions and complement the article's contributions to the advancement of knowledge of the object)

“To this end, educational methods need to be developed.” this idea appears isolated, show a greater relationship with the content of the abstract.

Thank you for pointing this out. We have revised the abstract, some of the background, and the discussion section entirely to clarify the discussion.

In the introduction, the objectives of the article, research questions should be evidenced. It is important to infer how this research contributes to the advancement of science, its contributions and social impact, identifying direct and indirect beneficiaries, as well as the relevance of this research within the framework of emerging paradigms that explain the educational dynamic.

To address your comments, we have revised the last paragraph of the background to ensure that the research question is appropriately guided.

The theoretical and empirical gaps of the situation under study should be evidenced to a greater degree, as well as the descriptors associated with the main variables studied; the variables and their dimensions should be described with greater emphasis. It is necessary to strengthen the citations of recent scientific literature that allow contrasting the descriptors of reality and show the state of knowledge related to the situation under study.

We conducted a qualitative study earlier with the same group of foreign subjects to establish our research questions. At the time of the initial submission of this paper, the paper summarizing the results was under review and therefore could not be cited. In this revised version, it is included in the list of cited references. In addition, we cited several previous studies that used similar questionnaires in studies examining difficulties in accessing medical facilities for foreign residents. We referred to those articles to confirm the validity of the variables studied. We have added a note about that in the text this time.

The situation under study, as well as the associated descriptors, the importance and relevance of the topic, the sense of contextualization in the region studied must be strengthened in their description; Likewise, the citation system of recent scientific literature related to the subject should be strengthened.

Thank you for your important comments. We have added to the text the meaning and validity of choosing Okinawa as the research field. We have also added the citations used as references in the preparation and analysis of this study.

It is important to propose a deductive route; it is necessary that the implications of the problem situation be described on a macro, meso and micro plane; situation that must be described in greater depth because it must be better argued; the problem must start from a more general scope, before falling into the variables of the investigation; that is, they should begin by presenting descriptors of the macro; then locate yourself in the meso and micro plane,

In order to respond to your comments as best we can, we have sequentially stated that this survey was conducted to understand the difficulties of foreign residents in accessing medical care and the assistance they need, that the actual survey is specific to senior foreign residents in Okinawa, that the institution conducting the survey is one university, and that the survey is limited to the satisfaction perspective. The Results and Discussion section has been modified to include a description of the subjects' overall satisfaction with Japanese healthcare first, followed by a description and discussion of the relationship between satisfaction and the individual survey items.

The scope of the applied survey should be evidenced to a greater degree as a basis for the generalization of the results.

Thank you for your valuable suggestions. In accordance with your remarks, we have discussed the significance of this study from a broader perspective in the Discussion and Conclusions.

I suggest reviewing the logic and systematization of the methodological design, for this I recommend considering the following elements:

It is necessary to identify and justify the typology of the article.  Is important  specify the predominant research approach, as well as the type of design. It would have to be determined if it applies: the inclusion and exclusion criteria in the sample must be specified.  Likewise, the information gathering techniques and instruments must be specified; instruments for each of the identified samples; criteria of validity and reliability of the instruments. Likewise, it is necessary to specify the information processing techniques to be obtained. The procedural systematization must be reviewed organize the research stages, in correspondence with the objectives of the article, type of research according to knowledge to be produced and expected products. Provide greater evidence of the operationalization of its variables, main dimensions and indicators that allow measuring the behavior of its fundamental variables.

Thank you very much for pointing out the multiple very important omissions in this paper. We have made additions and corrections to several pertinent sections of the paper. We hope you will take the time to review them.

The profile of the subjects that make up each group, their characteristics, is not clear; should systematize in a more organized way.

The issue of how to describe subject groups, etc. have been pointed out by reviewer 1. We have improved it as much as possible.

Systematize the methodology in a more organized, logical and coherent way

In response to your comments, we have added an explanation to the Methods section regarding the inclusion criteria of the subjects and how the survey items were selected.

Review the way to organize the results; there must be a better correspondence with the procedural systematization that is necessary to declare in the methodology, that is, how each component of the design leads to the different sections that are presented in the results. Review the correspondence between the methodological systematization and the systematization of the results, attend to variables, dimensions and indicators and their operationalization.

We have reviewed the entire paper to reflect the reviewers' suggestions to the extent that we can consider and revise them at this time. We would appreciate your review.

There is no analysis of the results, it is necessary to show: Organize the results in such a way that they are evident in relation to the objectives; there is no evidence of contrast between the objectives - supporting theory - meaning of the data itself - argument of the researchers.

We have made major additions to the discussion and considered as much as possible in the interpretation of the results.

Where is the argumentation or counter-argumentation with the supporting scientific literature?

In the Discussion section, we have examined several conflicting hypotheses, including interpretations of the relationship between dissatisfaction with communication and satisfaction with overall health care in Japan, and added some papers that contribute to our examination of the hypotheses.

It is necessary to strengthen the citations of recent scientific literature, citations of contrasting literature must be updated.

We have added several references that we have referred to in our own research planning and discussions.

the conclusions show correspondence with the identified objectives; demonstrate possibilities of generalization of the research to contexts with similar characteristics. Conclusions must transcend results. Conclusions should be more precise

We have reviewed the description in the conclusion and rewritten it in its entirety. We would appreciate your review.

I suggest organizing the conclusions in a more precise way, as conclusive ideas in correspondence with the objectives, variables and dimensions of the study, the results should be transcended.

Thank you for all the detailed and thought-provoking remarks you provided. In response to the above series of remarks, we have made some changes to the Background and Methods sections and completely revised the Discussion and Conclusions sections to clarify the discussion.

Once again thank you for reading our manuscript in detail.

Reviewer 3 Report

Authors present acceptance of healthcare professionals and their services in Japan, or strictly writing in Okinawa . Therefore that is the problem ,

Authors provided detailed analysis of that situation in Okinawa. Excellent, Ok, I agree. However, further authors present discussion on situation in Japan. My question is if the Okinawa is sufficiently representative to present the conclusions. May be , authors are right , ok, but it should be explained 

Beyon that I would ask authors what is the theoretical background of their studies. Organizational Human behaviour theories, or Human Resources Management? Authors focused on statistical methods , ok, but , who would be in practice the recipient of your results. I trust authors are able to answer.

Author Response

Authors provided detailed analysis of that situation in Okinawa. Excellent, Ok, I agree. However, further authors present discussion on situation in Japan. My question is if the Okinawa is sufficiently representative to present the conclusions. May be , authors are right , ok, but it should be explained

Thank you for reiterating a very important aspect of our paper. The Ryukyu Kingdom once flourished in Okinawa and developed its culture as a country distinct from Japan, which is evident in its food, music, and dialect. In the 19th century, the Ryukyu Kingdom was annexed by Japan and began to be governed homogeneously, with mainland Japan, as part of a centralized modern nation. However, Okinawa became a battleground during World War II and was occupied by the U.S. after the war. In 1972, Okinawa was reacquired by Japan; however, several U.S. military bases remained in Okinawa, causing a significant burden on the Okinawan people. The presence of U.S. military bases became the cause of inequality in Japan. Nevertheless, Okinawa is an international island that has actively introduced foreign cultures through the U.S. military bases. Hence, as you suggest, making generalizations about Japan, when we focus on the medical care in Okinawa, is something we must be careful about.

In contrast to Okinawa, the Japanese health care system is greatly uniform and controlled by the central government. Of course, this does not mean that the health care delivery system in underpopulated areas or isolated islands is identical to that of large cities, such as Tokyo and Osaka. However, the system allows patients to access almost the same quality of health care, irrespective of their residential location. The same is also true for medical care in Okinawa. On the main island of Okinawa, the medical care system is not very different from that in Tokyo.

Therefore, our response to Reviewer 3’s question is as follows: The survey conducted in Okinawa targeting foreign users of the health care system cannot be generalized for all of Japan. However, from the perspective of a medical system that accepts foreigners, we can obtain a degree of insight to improve the medical system in Japan as a whole.

We would appreciate your review of the following in the Limitations section of our discussion.

The ultimate goal of this study was to contribute to the inclusion of foreigners in health care. However, the actual study was a small-scale, exploratory study, conducted at a single university in Okinawa, where English is the official language, among a selected group of foreigners. Therefore, the generalizability of the results is severely limited.

Beyon that I would ask authors what is the theoretical background of their studies. Organizational Human behaviour theories, or Human Resources Management? Authors focused on statistical methods , ok, but , who would be in practice the recipient of your results. I trust authors are able to answer.

Thank you for pointing this out. Your question is very critical. It has made us think about the target audience of our paper and its context. Our paper addresses health care in Japan and Okinawa in the context of health care communication and policy, which may impact organizational behavior theories. However, the adoption of a public health ethics perspective for health care policy and health care communication is the theoretical feature of our paper. Foundational theories for health care policy are related to political philosophy. We are interested in the equality of health care delivery and advocacy for vulnerable populations. We believe that introducing a liberal, egalitarian perspective in medical communication and policy can improve medical communication overall. Although our paper is limited to Japan, and Okinawa, in particular, we hope to communicate with global readers who are interested in health care policy and communication through discussions on equality and advocacy for those who are vulnerable when it comes to health care.

Round 2

Reviewer 2 Report

Dear Authors.

Thank you for integrating the given suggestions into the manuscript.

Author Response

We thank you for taking the time to carefully review our paper. Our paper has already been checked by an English editing service provided by MDPI.
